# Semi-Supervised Learning for Forklift Activity Recognition from Controller Area Network (CAN) Signals

**DOI:** 10.3390/s22114170

**Published:** 2022-05-30

**Authors:** Kunru Chen, Thorsteinn Rögnvaldsson, Sławomir Nowaczyk, Sepideh Pashami, Emilia Johansson, Gustav Sternelöv

**Affiliations:** 1Center for Applied Intelligent Systems, Halmstad University, Kristian IV:s väg 3, 301 18 Halmstad, Sweden; thorsteinn.rognvaldsson@hh.se (T.R.); slawomir.nowaczyk@hh.se (S.N.); sepideh.pashami@hh.se (S.P.); 2Toyota Material Handling Manufacturing Sweden AB, Svarvargatan 8, 595 35 Mjolby, Sweden; emilia.johansson@toyota-industries.eu (E.J.); gustav.sternelov@toyota-industries.eu (G.S.)

**Keywords:** machine activity recognition, semi-supervised learning, learning representation, CAN signals, forklifts

## Abstract

Machine Activity Recognition (MAR) can be used to monitor manufacturing processes and find bottlenecks and potential for improvement in production. Several interesting results on MAR techniques have been produced in the last decade, but mostly on construction equipment. Forklift trucks, which are ubiquitous and highly important industrial machines, have been missing from the MAR research. This paper presents a data-driven method for forklift activity recognition that uses Controller Area Network (CAN) signals and semi-supervised learning (SSL). The SSL enables the utilization of large quantities of unlabeled operation data to build better classifiers; after a two-step post-processing, the recognition results achieve balanced accuracy of 88% for driving activities and 95% for load-handling activities on a hold-out data set. In terms of the Matthews correlation coefficient for five activity classes, the final score is 0.82, which is equal to the recognition results of two non-domain experts who use videos of the activities. A particular success is that context can be used to capture the transport of small weight loads that are not detected by the forklift’s built-in weight sensor.

## 1. Introduction

The forklift truck is a key piece of equipment in modern industry, and it is hard to imagine a production facility, warehouse, or logistics center without it. Each year, over one million new forklift trucks are sold worldwide [1]. The central role the forklift truck plays in industrial operation means that it is desirable to measure the activities of forklifts, to be able to monitor, analyze and improve the design of the material handling operation. However, forklift operators can perform their tasks in different ways, and detecting what operation is being performed is not trivial. Machine activity recognition (MAR) is a developing research field devoted to data-driven methods for recognizing the activities of equipment. The research in MAR is almost exclusively devoted to construction equipment (excavators, dumpers, haulers, etc.) and using external sensors such as accelerometers, microphones, and cameras. Forklift trucks have received far too little attention, considering their abundance in manufacturing operations. This is possibly because forklift trucks are considered “unrecognizable” equipment [2] with the external sensor approach due to their silent operation and the lack of highly articulated parts.

This paper presents an alternative approach to activity recognition for forklift trucks: one that does not build on external sensors but instead uses the internal Controller Area Network (CAN) data. The CAN bus protocol is an industrial standard for communication networks on vehicles. Since essentially every forklift truck manufacturer today uses CAN for internal communication, the approach presented in this paper should be widely applicable.

In the proposed approach, both labeled “laboratory” data and unlabeled real field operation data are used to design the activity recognition method. This is different from the typical MAR setup, which is to use supervised learning and data collected on 1–2 vehicles in one location, with a limited number of operators (often just one) who perform repeated tasks. Semi-supervised learning is employed to make use of the unlabeled field operation data and construct a classifier with higher accuracy than that solely from the small labeled “laboratory” data set. Combining this with expert knowledge in post-processing results in a high accuracy activity recognition system, as measured on a labeled out-of-sample data set.

The three contributions of this paper are as follows: (1) the first study on MAR for forklift trucks using CAN data, (2) the first demonstration of using unlabeled data with semi-supervised learning to build better classifiers in this domain, and (3) demonstrating successful capture of small weight loads that are not detected by the onboard load sensor but correctly identified from the activity context.

## 2. Literature Review

### 2.1. Machine Activity Recognition

Past MAR research has mostly focused on recognizing the activities of construction equipment. For example, the first attempt at using machine learning to recognize machine activities from sensor data was by Vachkov et al. [3], who used onboard data and self-organizing maps to recognize the actions of an excavator. They reported a recognition accuracy of about 93% for six operational modes: *loading bucket*, *moving load to a nearby truck*, *unloading bucket*, *return to the initial position*, *move bucket for leveling the load on the full truck*, and *idling*.

Several papers on construction equipment activity recognition have been published in the last decade. These are reviewed by Sherafat et al. [2], who categorize the approaches into three groups: (1) kinematic-based methods; (2) computer vision-based methods; and (3) audio-based methods. Kinematic-based methods use accelerometers, gyros, or control signals onboard the machines. Computer vision-based methods use cameras, and audio-based methods use microphones. Most of these methods do not employ factory-installed sensors on the machines but require additional sensors placed on or near the equipment. The studies on MAR for construction equipment typically aim to recognize 3–9 operation modes (including “trivial” modes, e.g., *engine off* and *idle*), achieving overall accuracies between 87% and 97% [2].

The categorization into three main approaches and the conclusions by Sherafat et al. [2] hold well also when considering papers that are not included in the review. Some exceptions are the work by Jung et al. [4], using movies of construction equipment downloaded from YouTube and two papers [5,6] which used unmanned aerial vehicles (UAV) to collect vision-based data without the constraint of fixed camera positions. Additionally, an early work by Vachkov et al. [3] and a recent work by Saari and Odelius [7] applied unsupervised learning techniques for MAR; whilst some research works started to use simulated data [8,9] and augment the data with known invariances [10]. However, apart from these research works, the MAR work has remained fully within the supervised learning paradigm, without using unlabeled data or synthetic data for the learning process.

There are very few publications on forklift truck activity recognition, and none report on performance measures. A German patent application [11] describes a system with ultrasound and motion sensors installed on the trucks and in the warehouse infrastructure where they operate. Alias et al. [12,13] suggest a solution with a few onboard sensors (not built-in) and cameras mounted in the warehouse infrastructure, to track forklift trucks and estimate the presence of load on the forks. In none of these cases are accuracies provided; the papers (and patent) describe possible solutions but no evaluations. Hence, the work presented in this paper is the first detailed description and evaluation of an activity recognition approach for forklift trucks.

### 2.2. Semi-Supervised Learning

Semi-supervised learning (SSL) is about making effective use of unlabeled data in learning. Chapelle et al. [14] provide a comprehensive introduction to the field and summarize the assumptions that form the basis for successful applications of SSL. Van Engelen and Hoos [15] present a review of the SSL field with a taxonomy of methods. The approach used for the forklift truck activity recognition fits within the framework of wrapper methods, where unlabeled data are incorporated via a pseudo-labeling step. In wrapper methods, a model is first constructed from labeled data using supervised learning, and unlabeled data are then labeled using this model. The most confident predictions are added to the labeled dataset, from which a new model is trained.

In our approach, we make two modifications to the pseudo-labeling step by (1) using all the pseudo-labeled data (instead of only using those with high prediction probabilities) and (2) learning new features (rather than training posterior models). This idea is similar to the naive semi-supervised approach presented by [16], where the authors show that deep neural networks are able to generalize well after training from noisy data.

## 3. Data Description

### 3.1. Data Collection

The data were sampled from internal CAN buses on forklift trucks using a compact CAN logger, a Vector GL1000, at a frequency of 10 Hz. Data were collected at two warehouse sites, one in Sweden and one in Norway, with the same machine type: reach forklift trucks with a load capacity of 1.6 tons. At the Swedish site, a camera was mounted above the dashboard and recorded the driver’s hand actions, which enabled later labeling of the activities. Two subsets of data were collected at the Swedish site with the same driver operating the same forklift; one with 58 min of data and the other with 27 min of data. The longer dataset represented normal operations, such as driving, picking orders, handling loads, and waiting to cross traffic. The shorter dataset focused more on demonstrating particularly complex activities, including various types of load handling operations and long periods of driving mixed with turns and stops. The Norway data were collected from a single forklift truck during two weeks of normal operation across different drivers. The Norway dataset was not labeled with activities.

In total, 262 signals were sampled from the CAN buses, with 14 of these signals recommended by domain experts as being particularly informative for recognizing forklift activities. The signals describe the steering command, wheel angles, fork reach command, fork reach position, lifting and lowering commands, fork height, load on the forks, engine speed, wheel speed, and heading of the truck.

### 3.2. Activity Labeling

At the center of MAR is the concept of “activity”. An activity is made up of a sequence of actions or events. Inherently, there are degrees of granularity or Level-of-Detail (LoD) in activities. For this paper, forklift experts were asked to propose activities that would be interesting to recognize and to sort them into desired LoDs. This resulted in a structure with four levels, presented in Figure 1. This structure is very similar to the LoDs for front-end loaders exemplified by Akhavian and Behzadan [17] or to the action hierarchy for construction machines suggested by Harichandran et al. [18]. The first level contains *engine off* and *engine on*. In the second level, while the *engine off* remains unchanged, *engine on* is further divided into *idle* and *active*. The *engine off* and *idle* activities can trivially be recognized, so the objectives of the data-driven method are the sub-activities of *active* in the deeper levels. In the next level, *active* is separated into three activities, i.e., *drive*, *load-handling*, and *other*. Finally, at the bottom of this hierarchy, while keeping *other* unchanged, *drive with load* and *drive without load* are stated as the sub-activities of *drive*; similarly, *take load* and *leave load* are the sub-activities for *load-handling*. It is worth noting that boundaries between activities are not always distinctive. Operators can perform two activities in parallel to maximize productivity, e.g., approaching the rack while lifting the fork. Thus, even human expert labels inherently exhibit uncertainty. Furthermore, the sequences of activities are expected to follow a logical order. For example, at Level 4, the four activities should appear in cycles: *drive without load*, *take load*, *drive with load* and *leave load*.

A forklift expert manually labeled each second of the two videos from the Swedish site by interpreting the drivers’ actions. This labeling was performed according to Level 4 in Figure 1. Figure 2 shows a comparison of the labeling on Levels 3 and 4 in these two datasets. Their distributions differ regarding the *other* category because the 27-min dataset does not reflect normal operation, while the 58-min dataset does. The *other* activity corresponds, e.g., to picking up orders and waiting for crossing traffic, which occurs very rarely in the shorter dataset. Despite this difference, the 58-min dataset was selected as the training set and the 27-min dataset for out-of-sample testing.

## 4. Methodology

The approach consists of the following five steps; each of them is described in detail throughout the subsequent subsections:1Train a baseline classifier using the labeled training data to be able to create pseudo-labels for the unlabeled data.2Train an autoencoder using the large unlabeled data set. The output of the bottleneck layer in this autoencoder is the autoencoder representation.3Fine-tune the autoencoder representation into a discriminative representation using the pseudo-labeled data.4Use the discriminative representation as input to a new classifier, trained on the labeled data.5Post-process the predictions from the classifier in step 4.

Figure 3 provides an overview pipeline of the five steps in the method and illustrates how the labeled, the unlabeled, and the pseudo-labeled data are used in different steps.

### 4.1. The Baseline Classifier

Sliding windows of size (K×M) were used as inputs for the baseline classifier (*K* is the number of signals, and *M* is the number of time steps). Following the work of Shi et al. [19], who also used CAN data (collected from displacement devices), logistic regression (LR), support vector machine (SVM), and random forest (RF) were tried in the baseline experiment. The best results were achieved with RF, and the baseline classifier is therefore denoted *baseline RF*.

### 4.2. Autoencoder

Deep autoencoders [20] with a fan-in architecture were used for the encoder and a symmetric fan-out structure for the decoder. The inputs to the autoencoder are the same as those used in the baseline classifier learning step (i.e., the K×M sliding time window). After training, the output of the bottleneck layer is a low-dimensional representation of the unlabeled data, denoted *auto_representation*.

### 4.3. Fine-Tuning Autoencoders into Auto_Discriminators

The *auto_representation* preserves the data variance, but this is not necessarily optimal for the activity recognition (it was verified in experiments that the autoencoder representation was not very effective for activity recognition). The autoencoder was, therefore, fine-tuned into a discriminative model by removing the decoder part and replacing it with a classification layer. We denote the result using *auto_discriminator*. Both the encoder part and the classification layer of the resulting network were then further trained using pseudo-labeled data. Pseudo-labeled data are the unlabeled data with labels predicted using the baseline classifier. Since this is a fine-tuning procedure, the network is not trained from scratch; instead, the weights in the encoder part are inherited from the autoencoder. The learning rate in this step is set to be very small to make only minor adjustments to the weights. The resulting representation is denoted by *auto_disc_representation*.

### 4.4. Post-Processing

Some activities are similar and easily confused, thus determining which one is being performed requires a context longer than two seconds. A two-step post-processing based on statistics and expert knowledge was employed to correct this. The first step is statistical and based on the probabilities of transitions between activities. This fixes intermittent spurious errors. The second step is an expert-based correction that builds on the expected logical order between operations.

#### 4.4.1. First Step: Transition Probability-Based Correction

The transition probability post-processing is designed to combine the prediction probabilities from the classifier with the conditional probabilities from the ground truth activity transition. For notation simplicity, prediction at time *t* is denoted by $\alpha_{i}^{t}$, where the index i∈{1,2,3,4,5} corresponds to the five target activities at level 4 in Figure 1, e.g., $\alpha_{5}$ is the prediction of *other*. For every time step *t*, the classifier outputs five prediction probabilities $y_{i}^{t}$, which can be interpreted as estimates of the conditional probabilities $y_{i}^{t}=P\left(\alpha_{i}|x^{t}\right)$, where $x^{t}$ is the signal input at time *t*. The a priori transition probabilities $P\left(\alpha_{i}^{t}|\alpha_{j}^{t-1}\right)$ are estimated from the 58-min labeled data. For example, $P\left(\alpha_{1}^{t}|\alpha_{4}^{t-1}\right)$ is the a priori probability that the activity switches from *leave load* to *drive without load*, before we have any information about the signal $x^{t}$. This probability is relatively small (approximately 5%) because most of the time the activity during the next second is the same as during the current second.

This post-processing adjusts the activity classification for time step *t* based on the activity in the previous time step t−1. It builds on the assumption that the previous activity is known, and the success of the post-processing depends on having some activity classifications that can be considered *certain*. A threshold θ is therefore set to select only the *uncertain* predictions as targets for the post-processing. If the highest prediction probability at time step *t* is larger or equal to θ, then no post-processing is applied to that prediction. Otherwise, the prediction is replaced by the modified value ${\hat{y_{i}}}^{t}=y_{i}^{t}\times P\left(\alpha_{i}|A\left(t-1\right)\right)$, where $A\left(t-1\right)\in \left\{\alpha_{1},\alpha_{2},\alpha_{3},\alpha_{4},\alpha_{5}\right\}$ denotes the *certain* activity at the previous time step. The activity for time *t* is then set to the activity with the largest ${\hat{y_{i}}}^{t}$, and the post-processing continues with the next time step, t+1.

#### 4.4.2. Second Step: Logical Order-Based Correction

The second post-processing step aims to fix longer sequences of erroneous predictions. There is an expected order between activities: *drive without load* should be followed by *take load*, which should be followed by *drive with load*, which should be followed by *leave load*, which should be followed by *drive without load*, and so on (see Figure 4). The activity *other* can follow or precede any activity.

Predicted activities that do not follow this expected logical order are considered wrong and are corrected to the most similar activity that agrees with the order. Note that “similar” means the activities that are one activity in Level 3 but two activities in Level 4 (see Figure 2): *take load* is similar to *leave load*, and *drive with load* is similar to *drive without load* (especially if the weight of the load is small, below sensor detection threshold).

### 4.5. Evaluation Metrics

The area under the receiver-operating curve (AUC) is considered gold standard when describing classification performance. However, when tests are conducted using one-against-all in a multi-class scenario, every test will be imbalanced, and our observation is that AUC tends to be overly optimistic whenever the classifier is good at recognizing one category. When evaluating imbalanced cases, it is advised to use balanced metrics [21], and we use the balanced accuracy (BA) for evaluating the one-against-all tests. The BA is defined as
(1)$$BA=\frac{TPR+TNR}{2},$$
where TPR and TNR are the true-positive rate and the true-negative rate, respectively.

When considering multi-class classification, the Matthews correlation coefficient (MCC) is recommended [22], and the MCC is therefore also reported for the full multi-class case. For the binary (two classes) case, MCC is defined as
(2)$$MCC=\frac{\mathrm{TP}\times \mathrm{TN}-\mathrm{FP}\times \mathrm{FN}}{\sqrt{\left(\mathrm{TP}+\mathrm{FP})(\mathrm{TP}+\mathrm{FN})(\mathrm{TN}+\mathrm{FP})(\mathrm{TN}+\mathrm{FN}\right)}},$$
where TP is the number of true positives, TN is the number of true negatives, FP is the number of false positives, and FN is the number of false negatives. Perfect prediction yields MCC = 1. Gorodkin’s generalization of MCC is used for the multi-class case [23]. It should be noted that BA and MCC can not be directly compared, since BA∈[0,1] and MCC∈[−1,1].

## 5. Results

The “original data” are the 14 signals suggested by experts to be useful for recognizing the target activities (see Section 3). Two-second-long snapshots were used, so the sliding window was of size 14×20, thus, the input was 280-dimensional. The activity label of each time window was determined by the last activity in that window snapshot. The window moved forward by 10 time steps (1 s) each time; therefore, the overlap between adjacent windows was 50%.

Supervised learning experiments were conducted using the 58-min labeled dataset as the training set, and 10-fold cross-validation was used to estimate variation in the results and determine hyperparameters. Two hold-out datasets were kept for testing: the shorter 27-min labeled dataset and 15.2 h of the unlabeled dataset. Instead of running the final models on the hold-out sets once, each model was trained in a stratified 10-fold cross-validation manner (after model selection), and all the resulting models were tested on the entirety of both hold-out sets. In this manner, all the results can be reported with mean and standard deviation, and a *t*-test can be used to estimate whether the differences in performances are significant.

### 5.1. Similarities between Data Sets

Autoencoders can be used to check whether the data occupy a low-dimensional manifold and if this manifold is approximately the same in different data sets. Autoencoder networks with different structures, varying in breadth and depth, were trained until achieving good performance of reconstructing the signals. After model selection, it was decided to use encoders with three hidden layers, with 128, 64, and 32 units, respectively. The decoder was symmetric in its structure and had three layers, translating the total autoencoder architecture into 280–128–64–32–N–32–64–128–280, where *N* denotes the number of units in the bottleneck layer. Activation functions in all the hidden layers were ReLu, except for the bottleneck units, which used a linear activation function. Backpropagation and early stopping were used for training. Figure 5 shows how the reconstruction error behaves depending on the number of bottleneck units (right) and the amount of data used for training (left).

Autoencoders were trained using unlabeled data from the Norway warehouse. The reconstruction error, when evaluated on hold-out test data, decreased when the amount of training data increased or when the number of bottleneck units increased. The behavior was essentially the same for data from the Norway and Sweden sites (see Figure 5). All networks in the right panel of Figure 5 were trained with 25 h of unlabeled data, and the autoencoders in the left panel all used three units in the bottleneck layer. The result shows that the low-dimensional manifolds where the data reside appear to be very similar between the Norway and the Sweden data sets, indicating that SSL should be applicable, since this is a requirement listed by Chapelle et al. [14]. Furthermore, being able to use much more data made a significant difference: an autoencoder trained with a large amount of unlabeled data from Norway was actually better at reconstructing the Swedish data than an autoencoder trained on all the available Swedish data itself.

### 5.2. Representations for Classification

The first hypothesis was that the sub-manifold representation from the autoencoder would, by itself, be useful for activity recognition. After all, principal component representations are often useful for classification tasks. This hypothesis was tested by constructing a number of RF classifiers, each using the bottleneck representation for one of the autoencoders in the right panel of Figure 5. The somewhat disappointing results from this experiment are shown in the left panel of Figure 6. None of the autoencoder representations (*auto_representations*) result in a classifier that is close in accuracy to the baseline RF model with raw data (cf. first column in Table 1). However, the classification improves with increasing bottleneck size. All the classification results in Figure 6 come from the evaluation on the 27-min hold-out test data from the Swedish site.

The discriminative representations were learned by fine-tuning the autoencoders as described in the methodology section (see Section 4.3). The same 25-h data as used for the autoencoder training were used for training the *auto_discriminators*, with pseudo-labels provided by the baseline RF classifier.

The right panel in Figure 6 shows the comparison between recognition and reconstruction performance (with RF classifiers constructed as above, except now using the discriminative representations). For the reconstruction results, new decoders with the same symmetric structure, i.e., N–32–64–128–280, were trained to reconstruct the unlabeled data from the *auto_disc_representations*.

The results in the right panel of Figure 6 show that the SSL method yields features that are good for discriminating between activities and that using few features is no worse than using many. A *t*-test was used to determine that there is no statistically significant difference between using low dimensional *auto_discriminators* with 3–6 features. Therefore, the three-dimensional setting was selected, which makes sense from Occam’s razor point of view and also provides the benefit of allowing visualization of the feature space. Consequently, the architecture of the *auto_discriminators* was chosen to be 280–128–64–32–3–5 (there are five activities in Level 4). All hidden layers used ReLu activation functions, and the output layer used a softmax.

The activity recognition results on the hold-out 27-min labeled test data with different representations (original space, *auto_representation*, and *auto_disc_representation*) are shown in Table 1. The three-dimensional *auto_disc_representation* provides activity recognition results that are not significantly worse than those of the baseline RF with the original 280 features and definitely much better than those with *auto_representation*. This result shows that the SSL method can map the original representation into a very low (almost 100 times smaller) dimensional representation where key information for recognizing forklift activities is preserved.

Figure 7 shows a visualization in the three-dimensional *auto_disc_representation* of the 58-min labeled data with Level 4 activity labels. The activities are quite well separated and the relationships between them are visible, e.g., *other* intersects with *drive without load*, *take load* is close to *leave load*, and so on.

### 5.3. Post-Processing

The first post-processed results we present in this section come from the RF trained on the *auto_disc_representation*. The decision threshold θ in step 1 is set to 0.8, a choice explained below. After the two-step post-processing, 27.5% of the predictions have changed, and 88.9% of these changes match the domain expert labels. Table 2 summarizes and compares the results of applying the two post-processing steps. The first column shows the recognition result without any post-processing. The two middle columns show the result after applying only one of the post-processing steps. The last column shows the result after both steps 1 and 2. It is clear that the first step of the post-processing, which cleans up spurious misclassifications, is necessary for the second (logic-based) step to be effective.

For testing the performance of the proposed method in terms of semi-supervised feature learning, post-processing was also applied to three other classifiers, i.e., the first three columns in Table 3. Except for the *baseline RF*, another RF classifier is trained with the pseudo-labeled large data set, namely the *pseudo-label RF*, and it is significantly different from the *baseline RF* that supplies the pseudo-labels. One more RF, an RF on top of *disc_representation*, is trained on top of a multi-layer perceptron (MLP). This MLP has the same network structure as *auto_discriminator* and is trained directly from the pseudo-labeled data without first training an autoencoder.

Figure 8 summarizes the classification results for the four methods for different values of the certainty threshold θ. Selecting the results corresponding to the highest MCC value for each method yields the results in Table 3. All classifiers trained with the large pseudo-labeled data set are approximately equal in performance, outperforming the *baseline RF*. This is because the *baseline RF* tends to give erroneous predictions with high prediction probabilities, which are then not fixed in the first post-processing step because the probabilities are higher than the threshold θ. A concrete example are the three consecutive predictions from the *baseline RF*: *take load*, *drive without load* and *take load*. The middle prediction, *drive without load*, is incorrect but not fixed with the first step due to having a high prediction probability. Then, the second post-processing step will change *drive without load* into *drive with load*, according to the first prediction of *take load*. Next, the last prediction of *take load* will be changed into *leave load* because its previous activity is now *drive with load*. If the spurious predictions are not corrected, then one wrong prediction can destroy several following prediction results (until the next *certain* and correct prediction is encountered).

Figure 9 illustrates the activity recognition results on the 27-min hold-out test set with and without post-processing. The activity recognition with the *auto_disc_representation* is quite good in itself but has problems with separating *take load* and *leave load*, and with detecting *drive with load* for light loads. The post-processing steps fix these confusions, and an “invisible load” is detected in a *driving* activity that occurs after about 1000 s. The load sensor is not sensitive enough to detect a load on the forks, but the activity recognition algorithm figures this out from the context, i.e., the prior recognized activities. Similar detection of “invisible loads” is also observed in the unlabeled data set, but no ground truth is available to verify their correctness.

### 5.4. Benchmark against Human Labeling

There is uncertainty in the expert labels because of the fuzzy boundaries between activities. It is therefore unrealistic to expect any classifier to recognize the activities with 100% accuracy. A test was conducted with two non-experts who were asked to label the forklift activities by observing movies from the data collection. Table 4 shows how the two non-experts match the expert labels on the 27-min hold-out test data from Sweden, measured with BA and MCC. When compared with the rightmost column in Table 2 or Table 3, it is striking how well the SSL method combined with post-processing is able to recognize the activities. It is only for the *other* activity that the non-experts perform significantly better.

### 5.5. Comparison against Motor Times

It was also checked how the recognized activities agreed with the “active motor time”, which represents the current industry standard measure for forklift utilization. The “active motor time” is robust and straightforward to compute, but not nearly as fine-grained and informative as the activity recognition. It is expected that the total time with active motors should correspond to the total time with activity for the forklift truck. A total of four time measurements are reported for each forklift truck: the time the drive motor has been active (*driving time*), the time the lift motor has been active (*lifting time*), the time either of the drive or lift motors have been active (*active time*), and the time none of the motors have been active (*inactive time*). Figure 10 shows how these match the recognized Level 4 activities on the 27-min hold-out test set. The *inactive time* matches well to the *other* activity, and the *active time* matches well to the sum of the remaining four activities. This confirms that the activity recognition results make sense.

The sum of *drive without load* and *drive with load* does not match the *driving time* perfectly, and the sum of *take load* and *leave load* does not match *lifting time* perfectly. This is to be expected since a load handling activity involves more than just using the lift motor (e.g., driving toward and away from a rack). Thus, the SSL method for predicting activities agrees with and improves upon the current method for measuring forklift utilization.

## 6. Summary and Conclusions

This paper describes and evaluates an activity recognition method for forklift trucks based on using streaming onboard CAN data. The method builds on first using a random forest classifier trained on a small data set collected in a laboratory, which is then used to label a large corpus of data from a warehouse in normal operation. Using this larger pseudo-labeled data set for training yields a more accurate classifier than the original one. Moreover, it is shown that this semi-supervised approach can be used to find a discriminative low-dimensional representation that allows visualization of the operational data with equally good accuracy, as compared to the original high-dimensional representation.

Furthermore, the results demonstrate that semi-supervised representation learning for MAR benefits from combining with two steps of post-processing; one statistical, considering transition probabilities between states, and one expert-driven, enforcing a strict “grammar” of how the activities are expected to arrive in a specific order. The final classifier is very accurate on a hold-out test set, with recognition accuracies of 88% for driving and 95% for load-handling activities. The Matthews correlation coefficient is 0.82. This is essentially as accurate as two non-experts who labeled the hold-out test data based on a video recording of the forklift truck activities.

The activity recognition results are evaluated quantitatively with standard classification metrics, as well as qualitatively by comparison with the conventional industrial approach. The comparison shows that the proposed activity recognition method represents a substantial improvement over the conventional method using “active motor time”, e.g., by detecting loads on the fork that are invisible to the onboard load sensor, and by providing counts and lengths of complete “load–transport–unload” cycles performed.

There are several potential challenges for future application of the proposed method, which will be the topic of further research. One is that the expert post-processing (the “grammar”) assumes a “strict” working cycle; activities after *leave load* are either *drive without load* or *other*. This may not always be the case; operators can perform two activities of *load-handling* at the same rack, i.e., *take load* can be performed immediately after *leave load*, without any driving in between. Other unusual scenarios are not considered in the method either, such as using the forks to push loads without lifting or lowering them. This was not encountered in the hold-out test set, which corresponds to “laboratory” type of data (the operators know that they are being monitored). As the results are extended to other warehouses under normal operation, there will likely appear new, creative ways to use forklift trucks. These challenges, however, are very likely not unique to forklift trucks and are also probably valid for human-operated construction equipment.

## Figures and Tables

**Figure 1 sensors-22-04170-f001:**
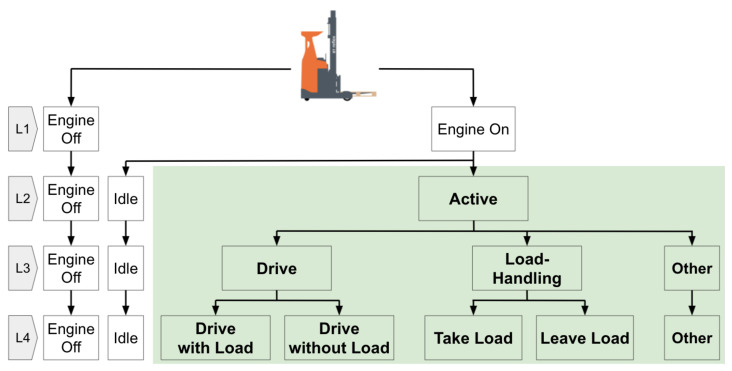
LoD of Forklift Truck Activities.

**Figure 2 sensors-22-04170-f002:**
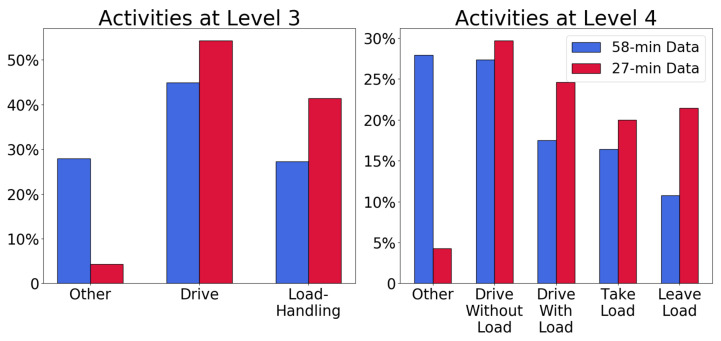
Distributions of activities at different Level-of-Detail, and for the two subsets of the Swedish dataset.

**Figure 3 sensors-22-04170-f003:**
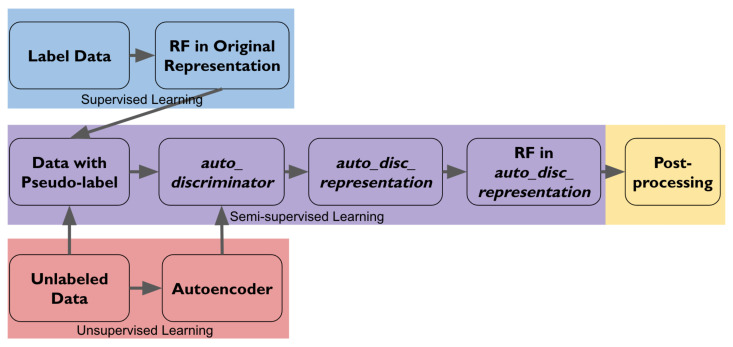
Pipeline of the method. Boxes in blue correspond to Section 4.1. Boxes in red correspond to Section 4.2. Boxes in purple correspond to Section 4.3. Box in yellow corresponds to Section 4.4.

**Figure 4 sensors-22-04170-f004:**
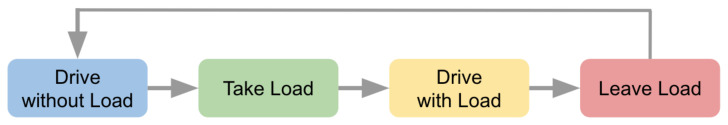
The logical order of forklift activities. Some papers [5,19] refer to this as “working cycles” of equipment.

**Figure 5 sensors-22-04170-f005:**
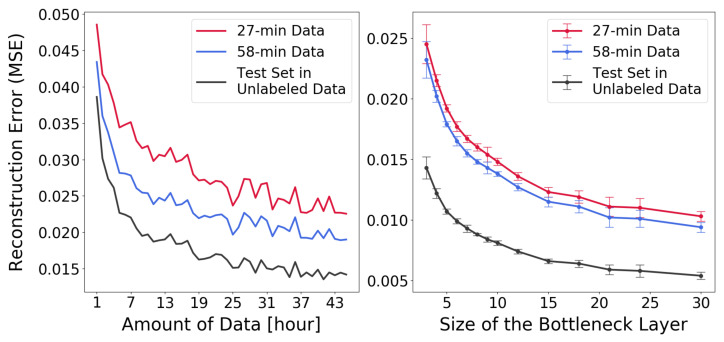
Reconstruction performance on the hold-out (unlabeled) test data from the Norway site and the two labeled datasets from the Swedish site, based on (**left panel**) adding more data during training and (**right panel**) increasing the bottleneck size.

**Figure 6 sensors-22-04170-f006:**
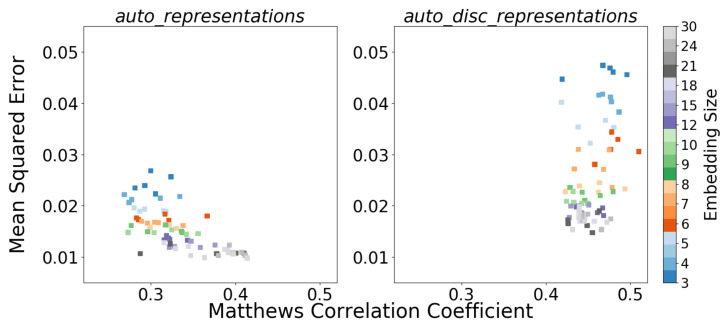
Comparison between recognition performances (in MCC) and reconstruction performances (in MSE) using (**left panel**) the *auto_representations* and (**right panel**) the *auto_disc_representations*.

**Figure 7 sensors-22-04170-f007:**
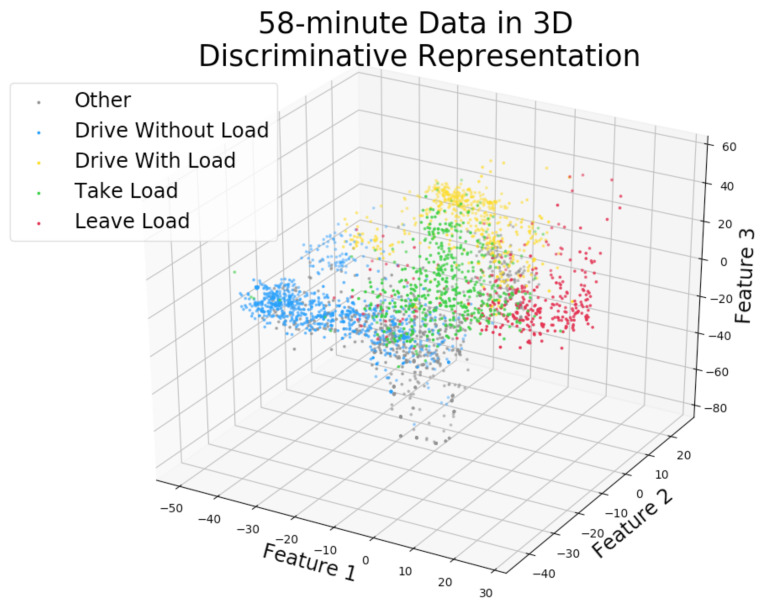
The labeled 58-min data (training set) from the Swedish site is shown in the discriminative 3-dimensional representation (activities are based on true labels from a human expert).

**Figure 8 sensors-22-04170-f008:**
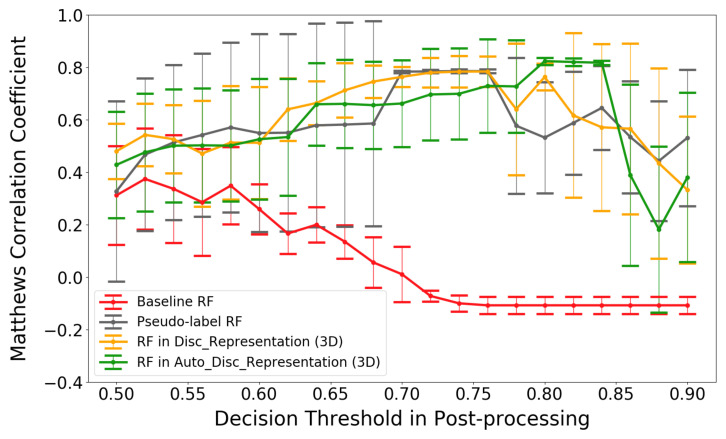
Classification results (using MCC) for post-processing with different decision thresholds. Error bars indicate standard deviations.

**Figure 9 sensors-22-04170-f009:**
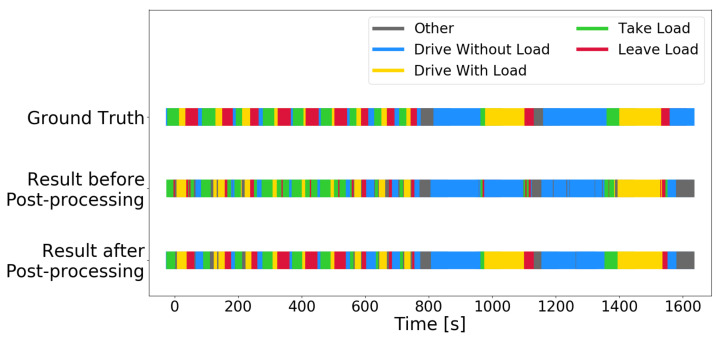
Ground truth and predictions (before and after post-processing) on each second in the 27-min hold-out test set.

**Figure 10 sensors-22-04170-f010:**
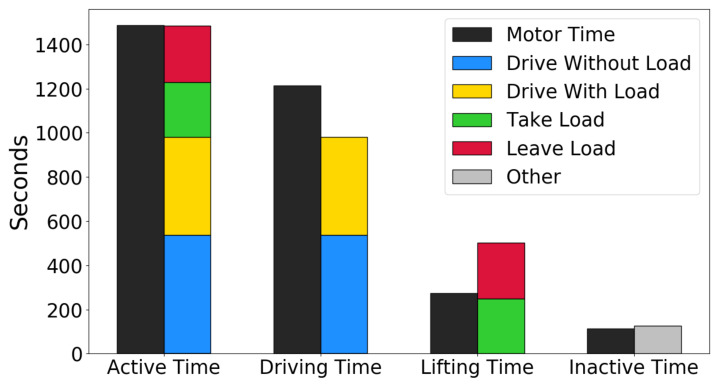
Predicted activities on the hold-out test set, compared to the current industry standard method (motor time) for measuring forklift utilization.

**Table 1 sensors-22-04170-t001:** Classification results of the RFs on top of different representations (mean ±2×std.)

		RF in Original Repre-Sentation (280D)	RF in *Auto_**Representation* (3D)	RF in *Auto_Disc_**Representation* (3D)
BA	Other	0.87±0.01	0.83±0.02	0.79±0.27
	Drive without Load	0.83±0.03	0.72±0.02	0.83±0.03
	Drive with Load	0.76±0.04	0.53±0.05	0.73±0.06
	Take Load	0.73±0.02	0.66±0.02	0.71±0.01
	Leave Load	0.62±0.06	0.59±0.02	0.59±0.03
MCC	All activities	0.52±0.04	0.30±0.03	0.47±0.05

Balanced accuracy BA∈[0,1] and Matthews Correlation Coefficient MCC∈[−1,1] are reported.

**Table 2 sensors-22-04170-t002:** Classification results of the RFs on top of different representations (mean ±2×std); boldface indicates the best performance (statistical significance) in each row.

		No Post-Processing	Step 1 Only	Step 2 Only	Two-Step (1 + 2)
BA	Other	0.79±0.27	0.85±0.17	0.79±0.27	0.79±0.25
	Drive without Load	0.83±0.03	0.87±0.03	0.61±0.20	0.94±0.02
	Drive with Load	0.73±0.06	0.77±0.01	0.60±0.26	0.95±0.01
	Take Load	0.71±0.01	0.74±0.02	0.74±0.13	0.88±0.05
	Leave Load	0.59±0.03	0.58±0.03	0.76±0.15	0.88±0.02
MCC	All activities	0.47±0.05	0.53±0.03	0.36±0.30	0.82±0.03

Balanced accuracy BA∈[0,1] and Matthews Correlation Coefficient MCC∈[−1,1] are reported.

**Table 3 sensors-22-04170-t003:** Classification Results from different experiments after post-processing (mean ±2×std); boldface indicates the best performance (statistical significance) in each row.

		Baseline RF (280D)	Pseudo-Label RF (280D)	RF in *Disc_**Representation* (3D)	RF in *Auto_Disc_**Representation* (3D)
BA	Other	0.92±0.01	0.87±0.00	0.88±0.03	0.79±0.25
	Drive without Load	0.65±0.36	0.92±0.01	0.90±0.08	0.94±0.02
	Drive with Load	0.62±0.40	0.93±0.01	0.87±0.15	0.95±0.01
	Take Load	0.76±0.19	0.85±0.01	0.84±0.04	0.88±0.05
	Leave Load	0.67±0.20	0.85±0.01	0.86±0.02	0.88±0.02
MCC	All activities	0.37±0.38	0.79±0.02	0.75±0.12	0.82±0.03

Balanced accuracy BA∈[0,1] and Matthews Correlation Coefficient MCC∈[−1,1] are reported.

**Table 4 sensors-22-04170-t004:** Activity recognition results of two non-experts on the hold-out test set (using video).

		Person 1	Person 2
BA	Other	0.90	0.83
	Drive without Load	0.93	0.93
	Drive with Load	0.94	0.93
	Take Load	0.89	0.89
	Leave Load	0.90	0.89
MCC	All activities	0.84	0.83

Balanced accuracy BA∈[0,1] and Matthews Correlation Coefficient MCC∈[−1,1] are reported.

## Data Availability

Data used in this research are provided by Toyota Material Handling Manufacturing, therefore, it is not open for public access.

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
