# Peer review of "Semi-Supervised Learning for Forklift Activity Recognition from Controller Area Network (CAN) Signals"

_sensors, 2022, doi:10.3390/s22114170_

Round 1

Reviewer 1 Report

This paper employs semi-supervised learning approach to handle Forklift activity recognition problem from CAN Signals. Here are my comments:

  1. There exists typographical oversight, e.g., the phenomenon of extra space for individual words.
  2. There are some grammatically incorrect sentences and must be corrected.
  3. The contrasts of information in the pictures of Results section are not intuitive and lack focus. The current ones look like just a pile of information, which is not all necessary.
  4. Punctuation marks are in the wrong positions for some sentences.
  5. Some definitions of parameters are not very accurate, and the layout is not clear either.
  6. The structure distribution of the algorithm is not clear, which is included in the method introduction content. It is suggested to list it separately.
  7. The content of the results section covers a wide range, with a slightly complex structure, and should highlight the merits of the method described in the paper.
  8. Some recent published highly cited papers are suggested to cite, e.g., An unsupervised fault diagnosis method for rolling bearing using STFT and generative neural networks, Journal of the Franklin Institute 357 (11), 7286-7707.

Author Response

Thank you for reading our paper and providing feedback. Here are our responses:

  1. We fixed all typographical errors we could find.
  2. We reviewed and corrected the grammar. 
  3. We tried to give a comprehensive view of our experiments, show different perspectives on the outcomes, and motivate our choices when moving forward. We have removed the discussion on the activity lengths since this is quite clear from other results. There are now six figures in the Result section. The first figure motivates the use of the unlabeled data by illustrating the overlap of embeddings. The second figure motivates using a discriminator representation instead of just the autoencoder and also motivates the bottleneck size. The third figure shows the resulting representation of the activities, illustrating that they separate quite well (it is, of course, much better if you can rotate it). The fourth figure shows the effect of an important parameter for post-processing. The last two figures illustrate how useful the results are in an implementation setting; does the solution capture important practical issues (“invisible” loads, length of transportation missions, and agreement with current state-of-the-art in industry to measure forklift truck usage)? We have improved the presentation (colors, symbols) in the figures. 
  4. Punctuation marks are fixed.
  5. We improved the language in Section 4.4.1 by re-arrangement and modification of the notation. 
  6. We re-phrased the first paragraph in “Methodology” to make the structure of the methodology easier to understand. Items in the list correspond to the subsections in Section 4 closely. 
  7. We have highlighted better the merits of our method, which is the use of unlabeled data and a “grammar” for the operations to get excellent performance on recognizing forklift activities (the first time ever this is demonstrated)  and the 3-dimensional representation that allows interacting with non-scientists on how the forklift truck moves between operations.
  8. We have added several more references to recent MAR papers.

Reviewer 2 Report

The presented paper talks about application of machine activity recognition (MAR) on forklift trucks.  It focuses on data-driven method for forklift activity recognition using Controller Area Network (CAN) signals and semi-supervised learning (SSL). 

The author claims that the work is one of the premier kind and there is no previous work on this subject. Which is also true from the initial search of the reviewer.  Furthermore the application is also very unique i.e. forklift machines so novelty is understandable.

The reviewer revisited the whole paper and the grammar (255 line where = were), figures and tables are all of good quality and presentable i.e. paper is well written and presented. 

I am completely satisfied with the novelty + curation of the paper and thus considers it accepted from my side. The only thing which is a bit less is the number of references. Maybe authors can increase the references e.g. FURBOT is an interesting forklift based autonomous vehicle and authors can cite them e.g. https://doi.org/10.3390/electronics9091370 . Rest the paper is well written and is ready for publication. 

Author Response

Thank you for your kind words on the manuscript. We did the following modifications according to your comments:

  1. We fixed typos and double-checked the grammar.
  2. More recent MAR papers were added in the literature review section.

Reviewer 3 Report

This paper was an interesting read and provided interesting approaches on tackling machine activity recognition problems.  The paper is well structured with organized writing. However, there are some confusing sections and missing details, which makes it difficult to read. In addition, this paper used the term “discriminator” extensively, which coincidentally similar to the discriminator network from the “Generative Adversarial Network (GAN)” but have very different meaning. I assumed that there are researchers that modify the GAN for semi-supervised learning, the discriminator network described in this paper did not match those network. First of all, there is no generative network to begin with, and the “discriminator” is not discriminating fake and real data. It seems like the author only added one layer of softmax layer for multi-class classification for the “ discriminator” definition. This method of taking the encoder network from auto-encoder and adding classification layer is called transferred learning instead of discriminator. Please provide supporting material to strengthen the proclaimed “discriminator”, or I would suggest the authors to change it into different name to prevent confusion and misinformation.

Line 116: 14 signals were used in this study, but there is no description on what are those 14 signals. Please describe what they are in details, including the type of signals, the range of its values, physical meaning instead of listing a few as examples. The description here is too vague. In addition, the CAN data was collected at 10 Hz, since CAN messages can be transmitted at different frequencies, was there any resampling or filtering done? If so, what they are and how was it done? Does the data logger provide those resampling?

Line 195: Typo, it should be “classifier” instead of “classier”

Line 192-207: This section discussed about the transition probability post-processing is somewhat confusing and hard to understand, it would be nice to take in-depth discussion on how and why this was done. Furthermore, adding diagrams or charts could help readers to understand this post-processing.  First, the classifier was mentioned to output five numbers , are they the softmax results from the classifier? Second, the transitional probabilities is denoted as , how was this value computed, and it was later shown with a similar probabilities, , are these two probabilities equivalent?  is defined as a set of states, which it is not consistent with .  Indices  were defined but  was not. If  is referred to the previous state, please mention it. Make sure to use consistent notation, and if needed, explain them well.

Figure 7. The figure is hard to read, the data are overlapping and the colors are too similar, suggest to change different data into different markers styles and colors, and change the error bars into dashed line or semi-transparent area for better differentiating from each other.

Line 318, Table 3: “disc_representation” is first mentioned, is it the same as the “auto_disc_representation” from earlier? Please define it or check consistency.

Table 3. It seems like after adding the 2-step post-processing, the baseline RF model is generally doing worse than without the post processing. It would be nice to include a small discussion on why this happened.

Line 352: Please show the formula or equation for the Earth Mover Distance.

Author Response

Thank you for providing very useful feedback. We did the following modifications according to your comments:

  1. To prevent confusion and misinformation, we changed the term “discriminator” into “auto_discriminator” and used italics wherever we used that term in our text. We chose to use “discriminator” in the first place because of the parallel to linear “discriminator analysis”. In linear analysis, principal components correspond to the autoencoder, and we see our tuning of the autoencoder into more discriminating features is similar to how linear discriminator analysis provides directions (dimensions) that are better for the discriminating than for the reconstruction task.
  2. We have added information on all the 14 signals in the text. The CAN signals were sampled at 50 Hz and then downsampled to 10 Hz. There was no specific filtering done.
  1. We fixed typos and double-checked the grammar.
  2. The post-processing section is modified, and more discussion, examples, and figures are added. The five numbers from the classifier are not always softmax results. For RF, these five numbers are prediction probabilities, calculated as the mean predicted class probabilities of the trees in the forest (where the class probability of a single tree is the fraction of samples of the same class in a leaf).
  3. The colors in Figure 8 (Figure 7 in the previous version) are improved for better visibility. We made the caps (the head and the tail of the error bar) clearer.
  4. We re-arranged the text around Table 3 (lines 338-343) so that “disc_representation” is easier to understand.
  5. We expanded a paragraph in Section 5.3 (starting with “Figure 8…”, lines 348-357) and added more discussion and explanation on the result from the baseline RF after post-processing.
  6. We removed the text about Earth Mover Distance (EMD) since we removed the two figures using this. This was done to shorten the Result section somewhat (as a response to a comment from Reviewer #1).

Round 2

Reviewer 1 Report

Although the authors have revised the paper, no clearly highlighted marks are made in the manuscript. Please made these marks in the revised version.

Author Response

Dear reviewer,

We attached a pdf file where the changes are marked with a yellow marker; please check.

Best Regards
